# Statistical Indicators of the Scientific Publications Importance: A Stochastic Model and Critical Look [†]

**Lev B. Klebanov [1,\*], Yulia V. Kuvaeva [2] and Zeev E. Volkovich [3]**

[1]   Department of Probability and Mathematical Statistics, Charles University, 116 36 Prague, Czech Republic
[2]   Department of Finance, Money Circulation and Credit, Ural State University of Economics,
     620144 Yekaterinburg, Russia; ykuvaeva1974@mail.ru
[3]   Software Engineering Department, ORT Braude College, Karmiel 21982, Israel; vlvolkov@braude.ac.il
[\*]   Correspondence: lev.klebanov@mff.cuni.cz
[†]   We dedicate this work to the blessed memory of Vladimir Mikhailovich Zolotarev, to whom we owe our
     interest in heavy-tailed distributions.

**Abstract:** A model of scientific citation distribution is given. We apply it to understand the role of the Hirsch index as an indicator of scientific publication importance in Mathematics and some related fields. The proposed model is based on a generalization of such well-known distributions as geometric and Sibuya laws. Real data analysis of the Hirsch index and corresponding citation numbers is given.

**Keywords:** citation distribution; Hirsch index; geometric distribution; Sibuya distribution

## 1. Introduction

In theory, a rather large number of indexes are proposed, which supposedly measure the significance of the scientific publications of an author. Among the most popular of them should be noted:

(i1)   the total number of citations of a particular author [1–3];
(i2)   Hirsch index of the author [4] (see also [5]).

It is these two indexes that we consider in the proposed work.
The definition of the numerical value of the index (i1) is clear from its name.
Recall the definition of the Hirsch index (see [4]). The Hirsch Index $h$ is the number of articles that have been cited at least $h$ times each. This index was introduced in [4], where its properties were explained. In our opinion, these do not correspond to the index purpose. However, we dwell on the description of both the positive and negative sides of the Hirsch index after constructing citation models for scientific articles. One of them has already been stated by us in preprint [6].

## 2. Citation Model Construction

We now turn to the construction of the author's citation model. It will be considered as a composite of two models. The first of it describes the process of publishing an article by one author which will be cited, and the second describes the process of citing such an article.
Let us make some assumptions, which we discuss later.

**Assumption 1.** *Let the probability of rejection or non citing of the manuscript be q and the decisions on publication of different manuscripts are taken independently.*

Then it is clear that the probability that the scientist will have exactly $k$ cited papers equals $q(1-q)^k$, $k = 0, 1, \ldots$. In other words, the number of publications of a scientist has a geometric distribution with parameter $q$. This distribution supposes that the number of an author publications may be arbitrarily large. However, $(1-q)^k$ tends to zero rather fast as $k \to \infty$ and, therefore, the mean value of the number of publications is not too large. The generating function of this distribution has the form

$$Q(z) = \frac{q}{1 - (1-q)z}. \tag{1}$$

Of course, here we assume that all the journals to which the author sends manuscripts have the same review system, i.e., all of them accept the manuscripts of this author with the same probability $1 - q$. More realistic is the situation with a random parameter $q$:

$$Q(z) = \int_0^1 \frac{q}{1 - (1-q)z} d\,\Xi(q),$$

where $\Xi$ is a probability distribution on $[0, 1]$ interval and then $\mathbb{P}\{X = k\} = \mathbb{E}(q(1-q)^n)$.

Let us go back to (1). How large may be the time spent by a scientist to publish a corresponding number of papers? Of course, this time is a random variable $T$ and we are interested in its distribution. The usual assumption on the working time is its exponential distribution with parameter $\lambda = \mathbb{E}T$ and the Laplace transform $\varphi(t) = 1/(1 + \lambda t)$. Suppose that times needed for the publication of $j$-th paper is $T_j$, and $T_1, T_2, \ldots$ are independent and identically distributed as $T$ random variables. Then the time needed for all publications has the Laplace transform

$$\sum_{k=1}^{\infty} \varphi^k(t) q(1-q)^{k-1} = \frac{1}{1 + \lambda t/q},$$

i.e., it has exponential distribution with the parameter $\lambda/q$.

It is natural to assume that each cited publication will produce some number of citations. Of course, the likelihood that the article will be quoted again depends on the number of previous citations.

**Assumption 2.** *Assume the probability that an article having $k - 1$ ($k \geq 1$) citations will not have new quotes equalling $p/k^\gamma$ where $p$ is the probability that the article will not be quoted for the first time. The parameter $\gamma$ is responsible for the speed of convergence of the rejection probability to zero.*

Consequently, the likelihood that the article will be quoted exactly k times equals $p/k^\gamma \prod_{j=1}^{k-1}(1 - p/j^\gamma)$. For the case of $\gamma = 1$, the generating probability function for the number of citations of this article is $1 - (1-z)^p$. The corresponding distribution function is named after Sibuya [7]. Below we consider the case of arbitrary positive $\gamma$. The corresponding study has general mathematical interest. Therefore, we provide it in a number of sections below.

## 3. Distribution of Citation Number of a Paper

Let us consider an ordered sequence of experiments $\{\mathcal{E}_n; \; n = 1, 2, \ldots\}$, where an event $A$ may appear in each of the experiments with the probability $p_n$. Define a random variable $X$ as the number of the first experiment in which $A$ appears. We suppose that $X$ is an improper random variable in the sense that it may take infinite value (that is, the event $A$ will never appear). For the case $\mathbb{P}\{X = \infty\} = 0$ we say that $X$ is a proper random variable. It is clear that, since we define any product from 1 to 0 to be 1,

$$\mathbb{P}\{X = n\} = p_n \cdot \prod_{k=1}^{n-1}(1 - p_k) \tag{2}$$

and

$$\mathbb{P}\{X = \infty\} = \lim_{n \to \infty} \prod_{k=1}^{n-1} (1 - p_k).$$

Particular cases are:

1. The probabilities $p_n = p$ are constant. So (2) is

$$\mathbb{P}\{X = n\} = p \cdot (1-p)^{n-1}, \quad \mathbb{P}\{X = \infty\} = 0 \tag{3}$$

corresponding to the classical geometric distribution. Its tail is

$$\mathbb{P}\{X \geq n\} = (1-p)^{n-1}, \quad m = 1, 2, \dots$$

Clearly, the tail and probabilities (3) decrease exponentially fast as $n$ tends to infinity.

2. The probabilities are given by $p_n = p/n$, where p is a number from the interval $(0,1)$. Equation (3) is transformed to

$$\mathbb{P}\{X = n\} = \frac{p}{n} \cdot \prod_{k=1}^{n-1}(1 - \frac{p}{k}). \tag{4}$$

According to (4) $X$ is a proper random variable and has, in this case, the Sibuya distribution with parameter $p \in (0,1)$ with the following tail

$$\mathbb{P}\{X \geq n\} = \frac{\Gamma(n-p)}{\Gamma(n) \cdot \Gamma(1-p)} \sim \frac{1}{\Gamma(1-p) \cdot n^p}$$

having heavy power asymptotic for $n \to \infty$. Such the distribution does not have a finite mean value. It is not difficult to see that

$$\mathbb{P}\{X = n\} \sim p/(n^{p+1} \cdot \Gamma(1-p)), \quad n \to \infty.$$

The presented distributions can be respected as a kind of "extreme points" from the perspective of the tail behavior for proper random variable $X$. Hence, it is natural to study roughly speaking the cases "happening between them"; namely to consider, for example, the situations when $p_n = p/n^\gamma$, with $p \in (0,1)$ and $\gamma > 0$. As it was mentioned above, the parameter $\gamma$ is responsible for the speed of convergence of the rejection probability to zero.

## 4. Main Result on Citation Number Distribution

The research subject is in the asymptotic behavior of the probabilities (2) for $p_n = p/n^\gamma$ with $\gamma \geq 0$. Additionally, to the discussed earlier values of $\gamma = 0$ or $\gamma = 1$, we distinguish the following two cases:

(A)　$0 < \gamma < 1$;
(B)　$\gamma > 1$.

Let us consider the case (A). We have

$$\mathbb{P}\{X = n\} = \frac{p}{n^\gamma} \cdot \prod_{k=1}^{n-1}(1 - \frac{p}{k^\gamma}). \tag{5}$$

Consider the product from right-hand-side of (5) in more details.

$$\prod_{k=1}^{n-1}(1 - \frac{p}{k^\gamma}) = \exp\left\{\sum_{k=1}^{n-1} \log(1 - p/k^\gamma)\right\} = \exp\left\{-\sum_{k=1}^{n-1}\sum_{j=1}^{\infty} \frac{p^j}{jk^{\gamma j}}\right\}$$

$$= \exp\left\{-\sum_{j=1}^{\infty} \frac{p^j}{j} \sum_{k=1}^{n-1} \frac{1}{k^{\gamma j}}\right\} = \exp\left\{-\sum_{j=1}^{[1/\gamma]+1} \frac{p^j}{j} \sum_{k=1}^{n-1} \frac{1}{k^{\gamma j}}\right\} \exp\left\{-\sum_{[1/\gamma]+1}^{\infty} \frac{p^j}{j} \sum_{k=1}^{n-1} \frac{1}{k^{\gamma j}}\right\}. \tag{6}$$

Here $[1/\gamma]$ is an integer part of $1/\gamma$. It is not difficult to see that

$$\exp\left\{-\sum_{[1/\gamma]+1}^{\infty} (p^j/j) \sum_{k=1}^{n-1} k^{-\gamma j}\right\}$$

has a finite positive limit as $n \to \infty$. This limit may depend on $p$ and $\gamma$. Let us denote it by $C_1 = C_1(\gamma, p)$. Therefore,

$$\prod_{k=1}^{n-1}\left(1 - \frac{p}{k^\gamma}\right) \sim C_1 \exp\left\{-\sum_{j=1}^{[1/\gamma]+1} \frac{p^j}{j} \sum_{k=1}^{n-1} \frac{1}{k^{\gamma j}}\right\} \quad \text{as} \quad n \to \infty. \tag{7}$$

Relations (5) and (7) give us

$$\mathbb{P}\{X = n\} \sim C_1 \cdot \frac{p}{n^\gamma} \cdot \exp\left\{-\sum_{j=1}^{[1/\gamma]+1} \frac{p^j}{j} \sum_{k=1}^{n-1} \frac{1}{k^{\gamma j}}\right\} \quad \text{as} \quad n \to \infty. \tag{8}$$

For $0 < \gamma j < 1$ the following asymptotic representation is known

$$\sum_{k=1}^{n-1} \frac{1}{k^{\gamma j}} = \frac{n^{1-\gamma j}}{1 - \gamma j} + \zeta(\gamma j) + o(1) \quad \text{as} \quad n \to \infty, \tag{9}$$

where $\zeta(u)$ is Riemann zeta function. Further considerations depend on properties of the number $\gamma$.

(i)　Suppose that $1/\gamma$ is not integer. Then $\gamma \cdot [1/\gamma] < 1$ and

$$\sum_{j=1}^{[1/\gamma]+1} \frac{p^j}{j} \sum_{k=1}^{n-1} \frac{1}{k^{\gamma j}} = \sum_{j=1}^{[1/\gamma]} \frac{n^{1-\gamma j}}{1 - \gamma j} \frac{p^j}{j} + \sum_{j=1}^{[1/\gamma]} \zeta(\gamma j) \frac{p^j}{j} + \frac{p^{[1/\gamma]+1}}{[1/\gamma] + 1} \sum_{k=1}^{n-1} \frac{1}{k^{\gamma([1/\gamma]+1)}} + o(1). \tag{10}$$

However, $\gamma([1/\gamma] + 1) > 1$ and, therefore,

$$\lim_{n\to\infty} \sum_{k=1}^{n-1} \frac{1}{k^{\gamma([1/\gamma]+1)}} = \sum_{k=1}^{\infty} \frac{1}{k^{\gamma([1/\gamma]+1)}} < \infty.$$

From this and (10) it follows

$$\mathbb{P}\{X = n\} \sim C_2 \cdot \frac{p}{n^\gamma} \cdot \exp\left\{\sum_{j=1}^{[1/\gamma]} \frac{n^{1-\gamma j}}{1 - \gamma j} \cdot \frac{p^j}{j}\right\}, \tag{11}$$

where $C_2$ depends on $p$ and $\gamma$ only.

(ii)　Suppose that $1/\gamma$ is positive integer. Then $\gamma[1/\gamma] = 1$ and

$$\sum_{j=1}^{[1/\gamma]+1} \frac{p^j}{j} \sum_{k=1}^{n-1} \frac{1}{k^{\gamma j}} = \sum_{j=1}^{[1/\gamma]-1} \frac{n^{1-\gamma j}}{1 - \gamma j} \frac{p^j}{j} + \sum_{j=1}^{[1/\gamma]-1} \zeta(\gamma j) \frac{p^j}{j} \tag{12}$$

$$+ \frac{p^{[1/\gamma]}}{[1/\gamma]} \sum_{k=1}^{n-1} \frac{1}{k} + \frac{p^{[1/\gamma]+1}}{[1/\gamma] + 1} \sum_{k=1}^{n-1} \frac{1}{k^2}.$$

It is known that

$$\lim_{n \to \infty} \sum_{k=1}^{n-1} \frac{1}{k^2} = \sum_{k=1}^{\infty} \frac{1}{k^2} < \infty$$

and

$$\sum_{k=1}^{n-1} \frac{1}{k} = \log(n) + \gamma_e + o(1),$$

where $\gamma_e$ is Euler's constant. Therefore,

$$\mathbb{P}\{X = n\} \sim C_3 \cdot \frac{p}{n^{\gamma + p^{[1/\gamma]}/[1/\gamma]}} \cdot \exp\left\{ \sum_{j=1}^{[1/\gamma]-1} \frac{n^{1-\gamma j}}{1 - \gamma j} \cdot \frac{p^j}{j} \right\} \quad \text{as } n \to \infty. \tag{13}$$

Now we see that the asymptotic behavior of the probability $\mathbb{P}\{X = n\}$ in the case A) is given by (11) and (13). From the relations (11) and (13) it follows

$$\mathbb{P}\{X = \infty\} = \lim_{n \to \infty} \prod_{k=1}^{n-1} (1 - p/k^\gamma) = 0,$$

so that $X$ is a proper random variable.

Denote by

$$b_m = \prod_{k=1}^{m-1} (1 - p/k^\gamma).$$

For the distribution tail $T_m$ we have

$$T_m = \sum_{n=m}^{\infty} \mathbb{P}\{X = n\} = (b_m - b_{m+1}) + \ldots + (b_s - b_{s+1}) + \ldots = b_m.$$

Particularly,

$$\sum_{n=1}^{\infty} \mathbb{P}\{X = n\} = 1.$$

If $1/\gamma$ is not a positive integer, then

$$T_m = \prod_{k=1}^{m-1} (1 - p/k^\gamma) \sim C_4 \cdot \exp\left\{ \sum_{j=1}^{[1/\gamma]} \frac{n^{1-\gamma j}}{1 - \gamma j} \cdot \frac{p^j}{j} \right\}, \quad \text{as} \quad n \to \infty, \tag{14}$$

where $C_4$ depends on $p$ and $\gamma$. Similarly, for the case of integer $1/\gamma$,

$$T_m \sim C_5 \cdot \frac{p}{n^{p^{[1/\gamma]}/[1/\gamma]}} \cdot \exp\left\{ \sum_{j=1}^{[1/\gamma]-1} \frac{n^{1-\gamma j}}{1 - \gamma j} \cdot \frac{p^j}{j} \right\} \quad \text{as } n \to \infty. \tag{15}$$

Let us consider the case (B). We have

$$\mathbb{P}\{X = n\} = \frac{p}{n^\gamma} \cdot \prod_{k=1}^{n-1} (1 - \frac{p}{k^\gamma}), \tag{16}$$

where $\gamma > 1$. Transform the product in the right-hand-side:

$$b_n = \prod_{k=1}^{n-1} (1 - \frac{p}{k^\gamma}) = \exp\left\{ \sum_{k=1}^{n-1} \log(1 - p/k^\gamma) \right\}$$

$$= \exp\left\{ -\sum_{j=1}^{\infty}\sum_{k=1}^{n-1} p^j/(jk^{\gamma j}) \right\} = \exp\left\{ -\sum_{k=1}^{n-1}\sum_{j=1}^{\infty} p^j/(jk^{\gamma j}) \right\}$$

$$= \exp\left\{ -\sum_{k=1}^{n-1} p/(k^{\gamma}-p) \right\} \overset{[}{n\to\infty]}\longrightarrow \exp\left\{ -\sum_{k=1}^{\infty} p/(k^{\gamma}-p) \right\}.$$

The series under an exponential sign converges because $\gamma > 1$. From latest relation we see that

$$\mathbb{P}\{X = \infty\} = \exp\left\{ -\sum_{k=1}^{\infty} p/(k^{\gamma}-p) \right\} > 0, \tag{17}$$

and $X$ is an improper random variable.

Therefore, for conditional probabilities we have

$$\mathbb{P}\{X = n | X < \infty\} \sim C_6 \frac{p}{n^{\gamma}} \quad \text{as} \quad n \to \infty, \tag{18}$$

where $C_6$ depends on $p$ and $\gamma$ only.

Summarizing, we obtain the following theorem

**Theorem 1.** *For the considered experiment scheme with probabilities given in (5) the following statements are true:*

- *If $\gamma = 0$ then $\mathbb{P}\{X = n\} = p(1-p)^{n-1}$, $n = 1, 2, \ldots$.*
- *If $0 < \gamma < 1$ and $1/\gamma$ is not a positive integer then*

$$\mathbb{P}\{X = n\} \sim C_2 \cdot \frac{p}{n^{\gamma}} \cdot \exp\left\{ -\sum_{j=1}^{[1/\gamma]} \frac{n^{1-\gamma j}}{1-\gamma j} \cdot \frac{p^j}{j} \right\} \text{ as } n \to \infty. \tag{19}$$

*If $0 < \gamma < 1$ and $1/\gamma$ is a positive integer then*

$$\mathbb{P}\{X = n\} \sim C_3 \cdot \frac{p}{n^{\gamma + p^{[1/\gamma]}/[1/\gamma]}} \cdot \exp\left\{ -\sum_{j=1}^{[1/\gamma]-1} \frac{n^{1-\gamma j}}{1-\gamma j} \cdot \frac{p^j}{j} \right\} \text{ as } n \to \infty. \tag{20}$$

- *If $\gamma = 1$ then*

$$\mathbb{P}\{X = n\} \sim p/(n^{p+1}\Gamma(1-p)), \quad n \to \infty. \tag{21}$$

- *If $\gamma > 1$ then*

$$\mathbb{P}\{X = n | X < \infty\} \sim C_4 \frac{p}{n^{\gamma}} \quad \text{as} \quad n \to \infty, \tag{22}$$

*and*

$$\mathbb{P}\{X = \infty\} = \exp\left\{ -\sum_{k=1}^{\infty} p/(k^{\gamma}-p) \right\} > 0, \tag{23}$$

*All $C, C_1 - C_6$ depend on parameters $p$ and $\gamma$ only.*

One of the reviewers of the first version of the paper advised us to study the form of the constants for some particular cases. We are very grateful him for the advice. Below we consider the case $\gamma \in (1/2, 1)$. In this case $[1/\gamma] = 1$ so that the sum under exponential sign in (19) contains only one summand. The calculations similar to give above leads to the following expression

$$\mathbb{P}\{X = n\} = \frac{p}{n^{\gamma}} \exp\left\{ -\frac{p}{1-\gamma} n^{1-\gamma} - \sum_{k=1}^{\infty} \frac{p^k}{k} \zeta(k\gamma) + o(1) \right\}.$$

In other words, the constant $C_2$ has form

$$C_2 = \exp\left\{-\sum_{k=1}^{\infty} \frac{p^k}{k}\zeta(k\gamma)\right\} > 0.$$

However, precise calculation of all other constant is rather difficult. We do not these constants for the aims of this paper and omit any other calculations of constants.

## 5. Comments

Theorem 1 shows that for $0 \le \gamma < 1$, the tail of the corresponding distribution is not heavy. Namely, the distribution has finite moments of all positive orders. However, the tail becomes heavier with growing $\gamma \in [0,1)$. In the case of $\gamma \in [0,1]$ the distribution is unimodal with mode equal to 1. For the values $\gamma \in [1,\infty)$, the distribution has a power-type tail, which is heavier than the ones occurring for $\gamma \in [0,1)$. In the case $\gamma \in [1,2)$ the conditional distribution under condition $X < \infty$ does not have the finite mean. However, for growing values of $\gamma \in [1,\infty)$ the tails of conditional distributions look to be less heavy. In the case of $\gamma \in [1,\infty)$ the conditional distribution has mode at 1.

## 6. The Case of Growing $p_n$

Above, we considered the case of the probability of event $A$ decreasing with increasing iment number. For completeness, consider the case of an increase of this probability.

Namely, suppose that in (1) $p_n = 1 - q/n^\gamma$ for $q \in (0,1)$ and $\gamma > 0$. Then

$$\mathbb{P}\{X = n\} = (1 - q/n^\gamma)\prod_{k=1}^{n-1}\frac{q}{k^\gamma} = \frac{q^{n-1}}{((n-1)!)^\gamma} - \frac{q^n}{(n!)^\gamma}. \tag{24}$$

It is clear that $\mathbb{P}\{X = \infty\} = 0$, and the tail of the distribution

$$T_m = \frac{q^{m-1}}{(\Gamma(m))^\gamma}$$

is a quickly decreasing function of $m$. Of course, distribution of $X$ has finite moments of all orders and it may have a mode not only at 1.

## 7. Back to the Distribution of Citation Number of One Author

We suppose now that the distribution of citation number of one paper has the form (5):

$$\mathbb{P}\{X = n\} = \frac{p}{n^\gamma}\cdot\prod_{k=1}^{n-1}\left(1 - \frac{p}{k^\gamma}\right), \quad n = 1, 2, \ldots$$

with $\gamma > 0$. Corresponding probability generating function is

$$\mathcal{P}(z) = \sum_{n=1}^{\infty} z^n \mathbb{P}\{X = n\}. \tag{25}$$

As was mentioned above, the number of cited paper is distributed according to geometric law with probability generating function (1):

$$Q(z) = \frac{q}{1 - (1-q)z}, \quad q \in (0,1).$$

The probability generating function of citation number of one author equals to the composition of $\mathcal{P}$ and $Q$, i.e., it is $\mathcal{P}(Q(z))$. It is clear that the tail of corresponding distribution is not heavy for $\gamma \in [0,1)$, it is heavy for $\gamma = 1$, and the distribution is improper for $\gamma > 1$.

Although the case of improper distribution seems to be not realistic, we discuss it for some particular cases below, after consideration of proper cases $\gamma \in [0, 1]$.

Let us remind that the case $\gamma \in (0, 1)$ leads to the light tailed distributions while $\gamma = 1$ leads to the laws with the heavy tail. The choice between models with light or heavy tails can only be made based on real data. Below we analyze some data of this kind.

### 7.1. Analyzing Data from Scholar Google "Mathematics"

Let us give the data for the part "Mathematics" on 16 February 2020 (see Table 1). The data given concern are the first 10 in the number of citations of authors. We do not give the names of these scientists. The table shows:

1. The serial number of the author;
2. The total number of citations by the author;
3. Hirsch Index;
4. The number of citations of the most popular work (By the most popular work we understand the work of this author having the largest number of citations among the works of this scientist);
5. Ratio of citations to squared Hirsch index;

**Table 1.** Citations "Mathematics".

| 1 | 2 | 3 | 4 | 5 |
|---|---|---|---|---|
| 1. | 448,557 | 270 | 28,303 | 6.15 |
| 2. | 162,457 | 98 | 44,406 | 16.92 |
| 3. | 159,123 | 147 | 26,929 | 7.36 |
| 4. | 138,820 | 64 | 110,393 | 33.89 |
| 5. | 101,662 | 59 | 35,640 | 29.20 |
| 6. | 99,206 | 78 | 41,647 | 16.31 |
| 7. | 85,288 | 59 | 55,293 | 24.50 |
| 8. | 84,918 | 48 | 18,901 | 36.86 |
| 9. | 77,319 | 98 | 11,715 | 8.05 |
| 10. | 73,989 | 72 | 17,153 | 14.27 |

Table 1 shows the first scientist has 2.76 times more citations than the second. In other words, the maximum of the observations is essentially greater than previous one. This observation leads us to think that the corresponding distribution has heavy tails (see [8,9]). As we have seen, it is possible for the case $\gamma = 1$ only.

### 7.2. Analyzing Data from Scholar Google "Biostatistics"

Let us give the data for the part "Biostatistics" on 16 February 2020 (see Table 2). The structure of Table 2 is the same as that of Table 1.

**Table 2.** Citations "Biostatistics".

| 1 | 2 | 3 | 4 | 5 |
|---|---|---|---|---|
| 1. | 478,691 | 227 | 66,611 | 9.29 |
| 2. | 301,786 | 132 | 59,613 | 17.32 |
| 3. | 253,221 | 208 | 26,127 | 5.85 |
| 4. | 223,038 | 218 | 10,184 | 4.69 |
| 5. | 199,143 | 169 | 23,447 | 6.97 |
| 6. | 178,855 | 117 | 39,271 | 13.07 |
| 7. | 150,695 | 105 | 42,485 | 13.67 |
| 8. | 119,199 | 111 | 20,666 | 9.67 |
| 9. | 108,648 | 140 | 20,842 | 5.54 |
| 10. | 100,491 | 111 | 30,315 | 8.16 |

Table 2 shows the first scientist has 1.59 times more citations than the second. Although it is it is less than the case of Table 1, the number is large enough to support our hypothesis on the presence of a heavy tail.

We do not give the data on the part "Statistics" but mention the situation is similar to that of the Tables 1 and 2.

### 7.3. Final Model for the Distribution of Citations

From the considerations of the two previous subsections, it follows that the most natural way to describe the distribution of citations is to choose $\gamma = 1$. This means

$$\mathcal{P}(z) = 1 - (1-z)^p, \quad Q(z) = \frac{q}{1 - (1-q)z}$$

and the probability generation function of citations distribution is given by

$$\mathcal{R}(z) = \mathcal{P}(Q(z)) = 1 - \left(1 - \frac{q}{1 - (1-q)z}\right)^p.$$

Denote by $Y$ the number of citations of a given scientist. It is clear that $\mathbb{P}\{Y = n\}$ may be found as the $n$-th coefficient of expansion $\mathcal{R}(z)$ in power series. We have

$$\mathcal{R}(z) = 1 - (1-q)^p(1-z)^p\left(1 - (1-q)z\right)^{-p}$$

$$= 1 - (1-q)^p \sum_{s=0}^{\infty}(-1)^s\left(\sum_{m=0}^{s}\binom{-p}{m}\binom{p}{s-m}(1-q)^m\right)z^s$$

$$= 1 - (1-q)^p + \sum_{s=1}^{\infty}(-1)^{s+1}\binom{p}{s}{}_2F_1(p, -s, 1+p-s, 1-q)z^s,$$

where ${}_2F_1$ is a hypergeometric function. Therefore,

$$
\begin{aligned}
\mathbb{P}\{Y = 0\} &= 1 - (1-q)^p; \\
\mathbb{P}\{Y = s\} &= (-1)^{s+1}\binom{p}{s}{}_2F_1(p, -s, 1+p-s, 1-q), \quad s = 1, 2, \ldots
\end{aligned}
\tag{26}
$$

It is possible to verify that $\mathbb{P}\{Y = 0\} > \mathbb{P}\{Y = 0\} > \mathbb{P}\{Y = s\}$ for all integers $s \geq 2$. Therefore, we meet a scientist without papers or with citing papers with maximal probability. If we limit ourselves by consideration of the scientists having at least one citation then the highest probability corresponds to authors with one citation.

The Laplace transform of the distribution of $Y$ has form

$$\mathcal{R}(e^{-t}) = 1 - \left(1 - \frac{q}{1 - (1-q)e^{-t}}\right)^p, \quad t \in [0, \infty).$$

Its asymptotic as $t \to 0$ is

$$1 - \mathcal{R}(e^{-t}) \sim \left(\frac{1-q}{q}\right)^p \cdot t^p, \quad \text{as} \quad t \to +0.
\tag{27}$$

This relation shows that the random variable $Y$ has moments of order less than $p$ and does not have moments of higher order. Because $p < 1$ the variable $Y$ has infinite mean. In practice, this means that some scholars have a very large number of citations. These citations refer to publications by a relatively small number of scholars. Of course, the data in Tables 1 and 2 are in agreement with these statements. It is important that the model is built on the assumption of the same capabilities of scientists. Even so, we must observe a greater variability in the number of citations of their publications.

Thus, the difference in the number of citations can be purely random and not say anything about the real contribution of the scientist into corresponding science field.

Of course, the proposed model is very idealistic, since it does not take into account the real difference in the capabilities of scientists, as well as in their equipping with the necessary tools and equipment. Taking into account the noted differences is likely to lead to the need to consider mixtures of the proposed distributions with different parameters $p$ and $q$. However, such a complication will not make it possible to distinguish scientists with a large contribution to science from those with a smaller impact.

Surely, the arguments presented for the choice of $\gamma = 1$ are rather crude, i.e., in reality, it may happen that $\gamma$ is close to unity. Although in this case, the distribution tail is not heavy, but over a very large (but finite) interval it is close to heavy. So, qualitatively, our conclusions will remain unchanged.

Based on the foregoing, we conclude that it is practically senseless to use the number of citations of a scientist's work to assess his contribution to science.

### 7.4. Remarks on the Model with $\gamma > 1$

In this subsection, we are trying to justify the possibility of using models with gamma greater than one. As already noted, in this model the probability $\mathbb{P}\{Y = \infty\}$ is not equal to zero. It is unlikely that this corresponds to the situation with the consideration of all scientists working in this field of science. However, a very long citation process (ideally, endless) is quite possible in the case of the most prominent scientists. For example, in the field of Mathematics, the works of Professor Andrei Nikolaevich Kolmogorov (1903–1987) continue to be cited. Over the past 15 years, they have been cited about 30,000 times, although more than 30 years have passed since the death of their author. It is highly probable that the citation process for these works will continue for a long time.

In addition, the concept of citation is somewhat arbitrary in our opinion. For example, in Mathematics, some theorems or other objects bear the names of scientists who were related to their preparation. Does the mention of these theorems and the corresponding names in some articles mean their citation? For example, many articles and books mention the Gaussian distribution without reference to the corresponding publication by Gauss. Is this mention a quotation? It seems to us that such kind of nominal results are not counted in determining the citation index. However, they certainly indicate the scientific significance of the result. It is very likely that for accounting for citations of this kind, models with a $\gamma$ greater than 1 may be required.

## 8. Hirsch Index

Recall that the definition of the Hirsch index was given on Page 1. Hirsch states that the proposed index $h$ is intended to rank authors of articles in the field of Physics. At the same time, it is noted that the index can be used in other fields of science. Since the number of citations is used in determining the index $h$, it seems plausible that $h$ is associated with this number. Hirsch notes that the number of citations is given by $N = \kappa h^2$. He wrote: "I find empirically that $\kappa$ ranges between 3 and 5" (We change notations of Hirsch. Namely, his *a* is our $\kappa$.). Further, Hirsch wrote: "$\kappa > 5$ is very atypical value".

Below we show that the Hirsch statements presented here are doubtful. In addition, the use of this index seems unreasonable.

Let's start by analyzing the data in Tables 1 and 2. Remind that the column 5 gives corresponding values of $\kappa$. Table 1 does not contain any $\kappa \leq 6$ while Table 2 has only one such value $\kappa = 4.69$. Other values of $\kappa$ are "very atypical", especially for Table 1. Table 2 contains 2 values of $\kappa \in (5, 6)$. Therefore, at least for such fields as "Mathematics" and "Biostatistics", Hirsch's conclusion about the "typical" form of proportionality between the number of citations of an author and the square of corresponding Hirsch's index seems to be incorrect. However, was Hirsch right in the field of "Physics"?

### 8.1. Data in "Physics"

Now we give the data on field "Physics", arranging them into a table in the same way as for Table 1.

Again, Table 3 has only one $\kappa \leq 5$, namely $\kappa = 4.88$. However, there are six values $\kappa \in (5, 6)$. The kappa values for the "Physics" area look smaller than for the "Biostatistics" area and significantly smaller than for the "Mathematics" area. The value of the Hirsch index for Physics has much less variability than for Biostatistics and Mathematics. The differences in citation numbers are much greater for Mathematics than in the case of Physics.

So, we see that Hirsch's understanding of the situation in Physics is closer to reality than in the case of Biostatistics and, especially, Mathematics.

*8.2. Data Comparison*

Continue the analysis of the data in Tables 1–3.

**Table 3.** Citations "Physics".

| 1 | 2 | 3 | 4 | 5 |
|---|---|---|---|---|
| 1. | 326,718 | 206 | 25,605 | 7.70 |
| 2. | 259,321 | 223 | 7275 | 5.21 |
| 3. | 240,376 | 200 | 15,651 | 6.01 |
| 4. | 232,057 | 206 | 26,535 | 5.47 |
| 5. | 231,746 | 218 | 15,589 | 4.88 |
| 6. | 227,530 | 206 | 15,684 | 5.36 |
| 7. | 217,495 | 144 | 35,746 | 10.49 |
| 8. | 200,565 | 191 | 11,807 | 5.50 |
| 9. | 198,735 | 190 | 7497 | 5.50 |
| 10. | 197,679 | 198 | 25,649 | 5.04 |

The average value of the Hirsch index in the case of Table 1 is 99.3 with a standard deviation of 66.45. The same indicators for Table 2 are 153.8 and 47.97, and for Table 3—198.2 and 21.73. We see that the standard deviation of the Hirsch index in the case of Mathematics is three times greater than in the case of Physics. On the contrary, the average value of the index is maximum in the case of Physics and minimum in the case of Mathematics. This shows that if Hirsch index is useful in the field of Physics, then its usefulness in the field of Mathematics is doubtful. Probably, it is true for Biostatistics too.

Authors with a higher Hirsch index are often inferior to others in the number of citations of the most popular works. For example, in Table 1, Author 1, having the highest Hirsch index, is inferior to Authors 2, 4, 5, 6 and 7 in the number of citations of the most popular work. In this case, Author 1 wrote his most cited work with co-authors, while author 2 did without co-authors.

It is clear that the Hirsch index does not exceed the number of cited publications of the author, which has an exponential distribution. Thus, the distribution of the Hirsch index has a light tail. Since the number of citations has a heavy tail, it is more variable than the Hirsch index. However, these two indicators are stochastically strongly related. Indeed, for the data in Table 1, the sample correlation coefficient between these indicators is $\rho 1 = 0.94$. On the other hand, the correlation coefficient between the Hirsch index and the number of citations of the most popular works is $\rho 2 = -0.23$. This coefficient indicates a small relationship between the indicators, and it is negative. In other words, a large Hirsch index is most likely not found among authors with highly cited individual articles. For Table 2, the values of the correlation coefficients equal to $\rho 1 = 0.702$, $\rho 2 = 0$, and for Table 3 $\rho 1 = 0.36$, $\rho 2 = -0.57$.

The increase in the Hirsch index with a decrease in the number of citations of the most popular work may result in the division of the work into a series of publications. However, when assessing the quality of a scientist's contribution, one should take into account that the publication of a series of articles instead of one may be caused not by a desire to increase the number of publications, but, for example, by a gradual insight into the essence of the problem under consideration. Such insight often requires a very long time, i.e., publication of a series of articles is justified. It should be noted that the publication of a series of articles naturally leads to an increase in the number of

self-citations. This increase cannot be considered as a flaw of the author and does not mean attempts to artificially increase the number of citations. At the same time, the presence of a series of publications (which increases the Hirsch index) cannot be considered as preferable to one highly cited work.

The presence of higher values of the Hirsch index in Physics compared to Mathematics can be explained by the use in modern Physics of expensive equipment in experimental Physics and/or the results obtained on it in theoretical Physics. Often this equipment is used by some laboratory or scientific group, and then transferred to another or others. After some time, this equipment again becomes available to the first group. Thus, new experimental facts arrive intermittently, and during the break they are processed and published. A theoretical analysis of the observed facts is also taking place. Then comes new information related to new experiments. Therefore, the very flow of information (both experimental and theoretical) contributes to the publication of not a single article, but a series of articles. This circumstance leads to an increase in the Hirsch index with a relative decrease in the number of citations of popular works.

A similar situation is absent in Pure Mathematics. Therefore, there the appearance of the series has much fewer reasons. Separate works appear, which often cover a substantial part of the problem under consideration. They cause a stream of citation of this particular work, and in a series of works. Thus, the Hirsch index becomes smaller than it would be if a series of articles were published instead of this one, but the most popular work causes more citations than each individual work in the series.

So, the use of the Hirsch's index has some basis in the field of Physics, but it is not related to what is happening in Mathematics.

For some areas of Applied Mathematics, a situation may be observed that is intermediate between what is happening in Physics and in Pure Mathematics.

However, it is not clear to us why not replace the Hirsch index with two. The first of these could be the number of all citations, and the second - the number of citations of the most popular work. The Hirsch index is stochastically quite closely linked to the number of all citations, so it and this number are "interchangeable". However, after the termination of the work of a scientist in a given field of science, the number of his publications does not increase and, therefore, the Hirsch index remains limited, while the number of citations can continue to grow unlimitedly. This is exactly what happens with the works of the most outstanding scientists of the past.

## 9. Distribution of the Hirsch Index

In this section, we obtain the probability distribution of the Hirsch index.

We introduce some notation. It is clear that the Hirsch index is a random variable. Let us denote it by $H$. We will denote the values of this $H$ by $h$. Our aim here is to determine the probabilities that $H = h$, i.e., $\mathbb{P}\{H = h\}$. In order for the event $H = h$ to occur, it is necessary and sufficient that:

(a)　no less than $h$ works were published;
(b)　$h$ of the published works are cited at least $h$ times, and the rest - less than $h$ times.

Suppose that $l$ works are published, and $l \geq h$. The probability of this event is $q(1 - q)^l$. Recall, the probability that a published work will be quoted $k$ times equals to $(p/k)\prod_{j=1}^{k-1}(1 - p/j)$. Therefore, the probability that the published work will be cited at least $h$ times equals to

$$\sum_{k=h}^{\infty} \frac{p}{k} \cdot \prod_{j=1}^{k-1}(1 - p/j) = \frac{\Gamma(h - p)}{\Gamma(h) \cdot \Gamma(1 - p)},$$

where $\Gamma$ is Euler gamma function.

The probability that a published work will be cited less than $h$ times is defined as

$$1 - \frac{\Gamma(h - p)}{\Gamma(h) \cdot \Gamma(1 - p)}.$$

Thus, the probability that $l$ papers are published, and the Hirsch index $H$ has taken the value $h$ is

$$q(1-q)^l \binom{l}{h} \cdot \left( \frac{\Gamma(h-p)}{\Gamma(h) \cdot \Gamma(1-p)} \right)^h \cdot \left( 1 - \frac{\Gamma(h-p)}{\Gamma(h) \cdot \Gamma(1-p)} \right)^{l-h}.$$

Now we see that

$$\mathbb{P}\{H = h\} = \sum_{l=h}^{\infty} q(1-q)^l \binom{l}{h} \cdot \left( \frac{\Gamma(h-p)}{\Gamma(h) \cdot \Gamma(1-p)} \right)^h \cdot \left( 1 - \frac{\Gamma(h-p)}{\Gamma(h) \cdot \Gamma(1-p)} \right)^{l-h}$$

$$= \left( \frac{\Gamma(h-p)}{\Gamma(h) \cdot \Gamma(1-p) - \Gamma(h-p)} \right)^h \cdot q \cdot \frac{\mu^h}{(1-\mu)^{h+1}},$$

where

$$\mu = \left( 1 - \frac{\Gamma(h-p)}{\Gamma(h) \cdot \Gamma(1-p)} \right) \cdot (1-q).$$

So, the random variable $H$ has the following distribution

$$\mathbb{P}\{H = h\} = (1-\nu) \cdot \nu^h,$$

where

$$\nu = \frac{(1-q)\Gamma(h-p)}{q\Gamma(h)\Gamma(1-p) + (1-q)\Gamma(h-p)}.$$

Note that this distribution is not geometric one because the value of $\nu$ depends on $h$.

Next, we are interested in estimating the tail of the distribution of $H$. To do this, we estimate the asymptotic behavior of the $\nu$. An application of the Stirling formula allows one to easily obtain that

$$\nu = \nu(h) \sim \frac{1-q}{q\Gamma(1-p)} \cdot \frac{1}{h^p}.$$

This formula immediately leads us to an asymptotic expression for the logarithm of probability $\mathbb{P}\{H = h\}$ for $h \to \infty$. Namely,

$$\log \mathbb{P}\{H = h\} \sim p \cdot h \cdot \log h, \quad h \to \infty.$$

It follows that the probability of the event $\{H = h\}$ decreases faster than the exponential function for $n \to \infty$. Of course, the tail of the distribution of $H$ also decreases faster than the exponential function. Therefore, there are moments of all orders of this distribution. Note that the distribution of the number of citations of articles by this author has an infinite mean value. So, if an author has a fairly large number of citations, then the ratio of the number of citations to the square of the Hirsch index can be arbitrarily large. This fact contradicts Hirsch's claim that $\kappa$ is bounded.

**Author Contributions:** Conceptualization, L.B.K.; investigation, L.B.K., Y.V.K. and Z.E.V. The authors have equally contributed to the writing, editing and style of the paper. All authors have read and agreed to the published version of the manuscript.

**Funding:** The study was partially supported by grant GAČR 19-04412S (Lev Klebanov).

**Conflicts of Interest:** The authors declare no conflict of interest.

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
