# Peer review of "Statistical Indicators of the Scientific Publications Importance: A Stochastic Model and Critical Look†"

_mathematics, doi:10.3390/math8050713_

Round 1

Reviewer 1 Report

1. I  am not  sure  if  Paretto  name  should  be  mentioned  in  the  Abstract.  Ahors  do  not  use  Paretto distribution, which  is  related  to  the  time  of  inpact of  a  publication over  next  publications  on  a  topic. Maybe  at  least  in  the introduction  they  could  make  a  note  on  this  fact.

2.   Their  Assumption 1   based  on  infinite  number  of  publication  by  an  author.  This  is  acceptable  for  a  mathematical  model.  But  seems  a  bit  unrealistic.   However,  I  agree just  for  the  purpose  of  model.  And  heavy  tail  distributions  behave  specifically  near  infinity.   And  grom  statistical  point  of  view,  you  know,   infinity   starts  above  30,  isn't it?

3.   In  Assumption 2  description  of  probability p  is  not  clear  enough (some  positive  number  0<p<1).  What  is  the  role  of parameter γ there? And  Sibuya  distribution  is also  needing  better  definition,  not  just  by  probability  generating  function (according  to  me,  this  is  a  discrette distribution).

4.  Further in Sections 3 to 7  authors  derive  some  properties  of  their  proposed distribution  depending on  values  of  proposed  parameters. I wonder,  why  do  they  analyze just  the  distribution  of  X (univariate r.v.,  and  not the  bivariate  distribution  of X  and  H (Hirsh index).  Maybe  this  is a  possible  next  project.

5.  Table  7.1  is  presented  as  Table 1  on p. 8.  Possibly  some  graphic  presentation of  these  data  may   be  source of  curiours discussions.  Same  remarks  a  valid  for  Tables 7.2  and  8.1.

6.   I could  not  find  any result that can  explain  the  statement  on  lines 17-19  at  the  beginning.  Maybe  authors add   some  points  about  this  in  their  conclusions..   Correlation  coefficients  can  be  calculated with  use  of  any  statistical  data,  but  theoretically  they  do  not  exist  when  marginal  distributions  do  not  have  second  moments. Nevertheless,  even  statistically  sorrelations  have  meaning  that  can  be  used.

Overal concusion:  This  is  a  good  article  which  offers  an  interesting  class of  probability  distributions, even  related  to  this  exotic, and  curious  for scientists  field.  I  recommend  this  article  for  publication. 

    It  is  up  to  authors  if  they  will take  in  consideration  my  reparks, or  not.  Better that  they  correct  the  minimum  of  the  details  I  noticed.

Author Response

Response to the Reviewer’s 1 review.

  1. Really, we do not mention Pareto name in connection to our model. This is because we were considering the situation where only discrete distributions were applicable. For example, we did not consider random time elapsed for papers publication. Looking at the first remark we decided to add a result on corresponding time distribution. However, it is not Pareto distribution in contradiction with Reviewer remark. Namely, it is geometric distribution. Corresponding result is given in the corrected version of our manuscript (see page 2). If the Pareto distribution would be the time distribution it will be catastrophic for our model. Really, it would mean that the mean time needed for the publication of corresponding number of the papers is very large and may be just infinite. It would make the model senseless. Basing on our result on the time distribution we did not mention Pareto name in Abstract.
  2. We agree with Reviewer remark. Therefore, we added the following (see page 2): “This distribution supposes that the number of an author publications may be arbitrary large. However, (1-q)k tends to zero rather fast as k->∞ and, therefore, the mean value of the number of publications is not too large”. And, of course, this Assumption is made for the mathematical convenience.
  3. Assumption 2 is now: “Assume the probability that an article having k - 1 (k >=1) citations will not have new quotes equals p/kgamma , where p is the probability that the article will not be quoted for the first time. The parameter gamma is responsible for the speed of convergence of the rejection probability to zero.” We add a citation on Sibuya’s paper. However, we did not change the definition of this distribution because it is defined by the probabilities of the values of corresponding random variable and by its probability generating function.
  4. It is not too easy to find bivariate distribution of X and H and we really plan to do this in another publication.
  5. Unfortunately, any graphic representation of the data from the Tables 1-3 seems to be not nice, because the citations numbers of the first author (in all tables) is essentially greater than for other authors. Therefore, the graph looks as one point situated far from the second point, which is not a point but the set of all others. By the way, it would be nice to have three-dimensional picture representing the number of citations, Hirsch index and the number of citations for most popular paper. Such picture looks just worth than in two-dimensional case.
  6. We agree with Reviewer here. Really, correlation coefficient for heavy-tailed distributions does not exist. However, its empirical counterpart exists. If the value of this coefficient is close to 1 then corresponding points are situated near a straight line. It shows high “empirical” dependence between corresponding coordinates (unfortunately, not between general random variables). We consider this fact as an intuitive support for the existence of dependence between X and H, say, but not as any proof of that dependence.

We are grateful to the Referee for his interesting and useful comments. They certainly contributed to the improvement of our article.

Reviewer 2 Report

The paper is OK but the language should be improved.

I have put a lot of remarks and comments in the paper. I have also put some math issues in the paper itself.

I attach it to this document

Author Response

Response to the Reviewer’s 2 review.

We are incredibly grateful to Reviewer 2 for the tremendous work to improve the English language in our article. All corrections are greatly appreciated. We made all the corrections of misprints in formulas pointed by the Reviewer (and it helps us to find some other misprints).

However, we did not accept another proof of formulae on the page 5 according to the following reasons. 1. Our proof is correct and consists in 3 lines. 2. The idea of it is clear. The proof, proposed by Reviewer, is a little bit shorter. It will take 2 lines. However, it is not our proof, and we see almost no reasons to prefer it.

On page 6, the reviewer proposed to calculate (estimate) the constants included in the corresponding asymptotic representations, at least for some cases. For gamma between ½ and 1 we calculated corresponding constant C2 as a function of p and gamma precisely. Corresponding expression is given now on the page 7 of the corrected manuscript. Calculation of other constant is possible as well, but corresponding functions of p and gamma appears to be rather bulky, and we do not give them in the paper especially because of qualitative character of our considerations.

Reviewer 3 Report

Comments to the Author

See the attachment

Author Response

Response to the Reviewer’s 3 review.

We are grateful to Reviewer 3 for his correction. They are greatly appreciated. As to Reviewer’s remark on the literature on indexes, we add citations for 3 publications by Garfield, E. and one for Hirsch, J.A.